# Do Hatchery-Reared Southern Pygmy Perch (*Nannoperca australis*) Develop Effective Survival Behaviour in a Soft-Release Site?

**DOI:** 10.3390/ani15182754

**Published:** 2025-09-21

**Authors:** James King, Peter Rose, Amina Price, Rafael Freire

**Affiliations:** 1Gulbali Institute, School of Agricultural, Environmental and Veterinary Sciences, Charles Sturt University, Albury, NSW 2640, Australia; james010king@gmail.com (J.K.); amprice@csu.edu.au (A.P.); 2North Central Catchment Management Authority, P.O. Box 18, Huntly, VIC 3551, Australia; peter.rose@nccma.vic.gov.au

**Keywords:** fish behaviour, development, restocking, in situ conservation, anti-predator behaviour

## Abstract

Captive breeding and release programmes are widely used to conserve threatened small-bodied freshwater fish, particularly in temperate Australian ecosystems, yet many of these fish lack the behavioural profile to survive in the wild. Placing fish into a soft-release site for a period to provide wild-like experiences with minimal threat to survival can improve post-release survival outcomes. Here, we compared the behaviour of Southern pygmy perch (*Nannoperca australis*) from a soft-release site with fish from a hatchery or the wild. In general, captive-bred fish behaved similarly to wild-caught fish, suggesting that behavioural patterns are largely inherent. However, we found that fish from the soft-release site were bigger and better at finding and using shelter. We also found subtle effects of the soft-release experience on anti-predator behaviour. We conclude that since survival behaviour appears to be mostly inherited in this species, soft-release programmes should focus on providing growth advantages and opportunities to learn shelter recognition and predator avoidance, as these specific benefits may be critical for survival after release.

## 1. Introduction

Freshwater habitats, including floodplains and wetlands, are under threat globally because of increasing water use, changes to salinity, and climate change [1]. In Australia, as in other parts of the world, small-bodied freshwater fish are highly vulnerable to a range of factors including habitat loss and loss of connectivity; flow regulation; climate change; changes to water quality, including sedimentation and nutrient run-off; and predation or competition from introduced species [2,3,4]. In response to biodiversity loss of small-bodied freshwater fish, captive breeding programmes have been suggested as a useful method to reintroduce or supplement existing populations to maintain or improve ecosystem health [5,6].

However, the success of released captive-bred animals is variable. Hatchery-raised fish frequently have low survival rates, with high mortality during the first few days to six months following release [7,8]. The success of captive breeding programmes and release is dictated by a range of factors, including the age or size of the fish at release; genetic history; rearing environment; location; timing and method of release; and number of fish released [4,6]. Behaviour is often the main line of defence against post-release mortality [9] and is well known to be a major driving factor in translocation success [10,11]. Life experiences and their timing during maturation can significantly impact a fish’s behaviour and its development [8,12]. Although the hatchery environment typically provides a safe environment, it rarely provides the experiences necessary for the development of natural behaviour [13,14,15]. Here, we consider natural behaviour as innate and learned actions and responses that an animal would typically exhibit in its native, wild environment.

The genetic background of captive-bred fish significantly influences behavioural development. Multi-generational captive breeding often results in artificial selection for traits that enhance performance in hatchery conditions but may be maladaptive in the wild [16]. However, first-generation (F1) fish from wild-caught broodstock typically retain wild-type genetic characteristics, although the hatchery environment may still fail to provide experiences necessary for optimal behavioural development.

“Soft release” is a broad term used to refer to many different practices depending on species and purpose, but, in general, soft-release practices aim to provide captive-bred animals with the experience necessary to develop natural behaviour. “Soft release” has been used to refer to a short period of acclimation at the final release site, such as acclimation to water temperature and flow [17]. Soft release may also involve longer-term exposure to semi-natural environments (e.g., closed pond systems, fenced off areas, artificial wetlands and dams), with the intention of exposing naïve animals to unfamiliar food sources, habitats, temperatures, and water quality [1]. For maximum effect, the soft-release site should be as similar to the final release site as possible, with consideration of features such as physical structures, food, other fish, and water quality.

However, the age of fish, duration of exposure, and, importantly, the genetic background of the fish are also important in determining post-release survival [8]. Additionally, as the soft-release site may have fewer or no predators compared to wild sites, the development of anti-predator behaviour may be limited. Extended exposure to natural-appearing environments that lack predation pressure could potentially reinforce inappropriate boldness or reduce vigilance when fish encounter similar habitats that contain predators. For F1 fish with intact wild-type genetics, the relative importance of behavioural conditioning versus other benefits (such as growth enhancement) in determining soft-release effectiveness remains unclear.

A useful framework to assess behaviour important for survival in freshwater fish is to consider behaviour along a spectrum of bold to shy [9]. This is particularly useful in this context because captive rearing typically shifts behaviour towards more bold traits that may threaten survival in the wild [18]. Bold/shy behaviour is usually examined in the laboratory using emergence, exploration, habitat choice, predator response, and novel food tests and can provide insights into the behavioural traits necessary for post-release survival [18,19].

Here, we compared the behaviour and physical development of Southern pygmy perch (*N. australis*; SPP) from three sources: captively bred (first generation) fish after six months in a soft-release site, captively bred (first generation) hatchery-only reared fish, and wild-caught individuals. SPP is a small-bodied freshwater fish typically found in slow-flowing or still waters with abundant aquatic vegetation, with a diet primarily consisting of small invertebrates and a relatively short lifespan, with reproduction occurring annually during spring and early summer [20]. This species is listed as near-threatened in the IUCN red list of threatened species and has important conservation applications since SPP (Murray-Darling lineage) are listed as endangered in New South Wales and vulnerable in Victoria [21], with a number of captive breeding programmes serving as insurance populations and to boost numbers of wild populations [22,23,24]. We predicted that SPP that had spent six months in a soft-release site would show “shyer” behaviour (i.e., a longer time to emerge from a hide, less exploration, more use of shelter and greater avoidance of predators, and novel food investigation) than same-age SPP from a hatchery. As key survival behavioural patterns are dictated by species and habitat, it cannot be assumed that particular bold/shy behaviour patterns are desirable, and for this reason, we also tested similar-age wild-caught SPP to provide a valuable reference point. We predicted that SPP from a soft-release site would show similar behaviour to wild-caught fish. Lastly, we also tested a fourth group, the “Juvenile” group, comprising captive-reared SPP from the same cohort as other captive-reared fish but at 3 months of age (i.e., before assignment to other groups). Our reason for testing this fourth group was to examine if behaviour changed drastically towards “bolder” traits during six months of captivity. However, due to the differences in size and water temperature, comparison with the Juvenile group was limited, but we have included this latter group in our report for completeness.

## 2. Materials and Methods

### 2.1. Subjects and Source of Fish

All procedures were approved by Charles Sturt University’s Animal Ethics Committee (A23842). We tested the behaviour of 118 Southern pygmy perch (*N. australis*) from four different groups. All captive-bred fish were of the F1 generation and from wild-caught broodstock that had been maintained in captivity for 3 years in a Recirculating Aquaculture System (RAS) at Middle Creek Farm (Stratford, Victoria, Australia) and fed pellets daily and live food at least weekly. “Hatchery” fish (N = 30, mean length 43.3 ± 1.0 mm; 17 females and 13 males) were kept at Middle Creek Farm until 9 months of age. “Soft release” fish (N = 27, mean length 57.9 ± 1.0 mm; 10 females and 17 males) were from the same F1 cohort as hatchery fish but placed in a man-made wetland at Strathfieldsaye (Victoria, Australia, Appendix A) at 3 months of age for six months of soft-release experience. This site is adjacent to wild Southern pygmy perch habitat but lacks invasive competitors (e.g., *Gambusia holbrooki*) present in nearby wild sites. The soft-release site was a man-made pond, approximately 12 m in diameter and 1.2 m deep at its deepest point. The pond contained a range of aquatic vegetation and emergent macrophytes. “Wild” fish (N = 30, 52.8 ± 1.0 mm; 12 females and 18 males) of around 9 months of age based on timing of collection and size were collected from a creek near the soft-release site (Appendix A) to provide ecological reference for natural behaviour. While these fish represent different genetic lineages than the captive-bred fish, they serve as important benchmarks for regionally appropriate behaviour. Wild and Soft-release fish were both collected in June 2024 using a combination of backpack electrofishing and rectangular 45 × 25 × 25 cm bait traps with a 4 cm opening. We used a Smith-Root electrofishing backpack unit (Smith-Root LR-20B, Vancouver, WA, USA) to apply the minimum electrical current to attract and stun fish, using eight standardised 150 s intermittent electrofishing shots. The setting on the unit was varied to account for changes in water conductivity. Immobilised Southern pygmy perch were collected with a dip-net with a 3 mm mesh.

“Juvenile” fish (N = 31, unsexed) from the same F1 genetic stock as Hatchery and Soft-release fish were tested at 3 months of age to provide developmental context, though temperature differences in testing limit direct comparisons with adult groups (see the Section 2.2 below).

### 2.2. Laboratory Housing

All fish were transported in aerated tanks to our behavioural testing laboratory in Albury (New South Wales). All fish were kept in an RAS (total volume of 1000 L) with a moving bed biological filter to manage nitrous waste and UV treatment to control infectious agents. Fish were kept in groups of 12 to 20 in square 35 L aquariums, with a 15 cm piece of PVC pipe to provide enrichment. All fish were kept at a dissolved oxygen level of 9–10 mg/L and a pH 6.5–7.5, with a 12 h day length. Water quality was monitored daily using testing kits for ammonia, nitrite, nitrate, and alkalinity (API freshwater testing kits, Mars Inc., Chicago, IL, USA), and a Horiba U-52 Multiparameter Water Quality Meter used to measure other parameters (temperature, dissolved oxygen, conductivity, pH). Fish were fed defrosted and decapsulated brine shrimp eggs to satiation at the end of each day and uneaten food was removed the following morning.

To match ecological conditions, Juvenile fish were maintained at 21–22 °C, reflecting the warmer temperatures typical of their developmental stage, while adult groups were maintained at 13–14 °C, consistent with seasonal conditions at both the hatchery and collection sites. Adjusting juvenile temperatures downward would not have reflected their natural thermal environment at that age and could have suppressed normal development of behaviour. Conversely, raising the temperature for adult fish would have required a prolonged acclimation period to avoid thermal stress, which could itself have influenced behaviour. As such, while this temperature protocol prioritised ecological relevance for each group, it introduces a limitation: direct statistical comparisons between Juvenile and adult groups should be interpreted with caution due to the confounding effects of water temperature.

### 2.3. Behavioural Testing

Behavioural tests were undertaken in two separate tanks that were connected to the holding aquariums to provide the same water quality. To minimise handling and disturbance tests on emergence, exploration, habitat choice and response to a model bird predator were performed sequentially, on all fish and in this order, in a plus maze made of white acrylic (Figure 1a and Appendix A), similar to the apparatus used by Freire et al. [25]. The fish predator test was carried out in a rectangular tank (Figure 1b). Behaviour was recorded from outside the testing room using overhead video cameras (Dahua 5231 Startlight, Dahua Technology Pty Ltd., Melbourne, Australia).

To test emergence behaviour, fish were confined to the refuge area (Figure 1a) for 10 min for acclimation with the guillotine doors to arms A2, A3, and A4 removed. An acclimating period of 10 min was sufficient for behaviour to return to normal after moving to this tank. The dividing guillotine door in A1 was then raised by 10 cm, and the time taken (latency) for the whole fish to emerge from the refuge was measured. Fish that failed to emerge after 10 min were scored as “not emerged” and gently ushered out of the refuge area using the handle of a net to begin the next test.

The exploration test commenced immediately on exiting the refuge, where exploration was considered the movement of the fish throughout the entire plus apparatus. The number of arms entered over a 10 min period was recorded.

To test habitat choice fish were confined to the centre area of the plus apparatus for a period of 5 min to acclimate. Since the fish were not netted, we found that a 5-min acclimating period was sufficient for behaviour to return to normal after confinement. Each arm of the plus apparatus had a substrate randomly allocated to pre-determined positions: small rocks (approx. 2 mm), artificial plants, large rocks (approx. 15 mm), and empty (Appendix A). After acclimation, the fish was released from the centre of the apparatus and able to swim freely between the four arms. Use of a basket to contain the substrate inadvertently provided fish with a ledge to shelter under. Instantaneous behavioural sampling was used to record the location of the fish each minute for 10 min, including if the fish was under a basket ledge.

### 2.4. Predator Tests

Following the habitat choice test, the fish was placed in the 10 cm area at the top of arm 3 for 5 min to acclimate (Figure 1a). A taxidermy piscivorous bird (Nankeen Night Heron, *Nycticorax caledonicus*) was placed outside of the plus apparatus (Appendix A), and arms 2 and 4 blocked using the guillotine doors to limit flight directions and ensure constant view of the model bird. After acclimation, the guillotine door was removed, and the bird model was made to swivel 90° every 4 s using a mechanical stand, providing view of the model bird to the SPP. The initial response to the predator—no response (fish remaining in place but with visible fin movement or movement towards or parallel to the predator), freeze (when the fish remained in location with no visible movement for more than 1 s) or avoidance (moving away from the predator)—was recorded.

After a minimum 12 h recovery period, the fish was placed in a rectangular tank (Figure 1b) containing a live, 25 cm long, piscivorous Murray cod (*Maccullochella peelii*). The SPP was initially contained within a refuge area for 10 min to acclimate, with an opaque barrier blocking view to the predator. The opaque barrier was raised, permitting view of the predator through a solid transparent barrier, and the refuge barrier was removed to give the SPP access to the rest of the apparatus. The initial response to the predator (i.e., no response, freeze, or avoid) was recorded.

### 2.5. Novel Food Test

Novel food tests were undertaken on individual fish in the 35 L aquariums following food deprivation for 24 h. Frozen bloodworms were placed on the centre bottom of the tank for 60 min. The time from presentation to investigation of the food was recorded. Investigation was defined as approximately less than 5 mm, based on an experimenter estimate, to the food for more than 2 s in a feeding (nose-down) position.

Immediately after the last behavioural test, Hatchery, Soft release, and Wild fish were euthanised by Aqui-S (Huber group, Lower Hutt, New Zealand) overdose at a ratio of 3 mL:10 L water, and weight, length, and sex were determined. Juvenile fish were not euthanised because they were able to be re-homed.

### 2.6. Statistical Analysis

Statistical analysis was conducted using R Statistical Software [26]. Fish weight and length met assumptions for parametric analysis and were analysed using a General Linear Model and a post hoc Tukey’s test to identify differences between groups. Weight and length analysis included three groups (Hatchery, Soft release, and Wild fish) as Juveniles were a different age and clearly different in size. A Fulton condition factor was calculated for the three adult groups of fish using the formula K = 100 × Weight/Length^3^ [27]. Other variables (emergence, arm exploration, habitat choice, and predator responses) were not normally distributed according to the Shapiro-Wild tests, and transformations to allow parametric tests were not successful. These latter variables were therefore analysed using Kruskal–Wallis non-parametric tests, followed by Dunn’s test with a Bonferroni correction for pairwise post hoc comparisons. Counts of the number of times fish were recorded in each habitat were converted to percentage of residence time in each habitat for presentation of the results. Binomial variables—use of the ledge (i.e., used ledge, did not use ledge), freeze or avoid responses to either the avian predator or fish predator (i.e., did freeze, did not freeze; avoided, did not avoid), and novel food (i.e., inspection, no inspection)—were compared between groups using Generalised Linear Models. Primary analyses focused on comparisons between F1 Hatchery and Soft release fish from identical genetic stock, with Wild fish providing ecological reference and Juvenile fish providing developmental context.

## 3. Results

### 3.1. Fish Size

Soft release fish were significantly larger than Hatchery fish in both weight (mean difference 1.6 g; Table 1; Figure 2a) and length (mean difference = 14.59 mm; Table 1; Figure 2b), representing approximately 20–30% size advantages (both *p* < 0.001). Soft release fish had a Fulton condition score of 1.33 ± 0.3, compared to 1.25 ± 0.03 for Hatchery fish and 1.26 ± 0.03 for Wild fish.

### 3.2. Behaviour

Although Soft release fish emerged sooner from the refuge (73.3 s) than other fish (Appendix A), the time taken to emerge was not significantly different between the four groups (Table 1). In the exploration test, there was no significant difference in the number of arms explored between the three groups of adult fish (Table 1; Figure 3).

In the habitat choice test, in general, fish showed a preference for any arm with a habitat over no habitat, with small rocks and plants being used most often and the arm with no habitat (Empty) being used least often (Table 2). However, there was no significance difference between groups in the frequency of use of each habitat (Table 2).

Use of the ledge was significantly influenced by group (Table 1), with Soft release (Mdn = 1, IQR = 0) and Wild fish (Mdn = 1, IQR = 1) using a ledge significantly more than Hatchery fish (Mdn = 0, IQR = 1; *p* < 0.001; Figure 4).

### 3.3. Predator Tests

Most fish showed at least one type of anti-predator response (Appendix A). There was no significant difference between groups in whether an avoid or freeze response was demonstrated in response to either the model avian predator or the fish predator (Table 1).

Of the fish that demonstrated an avoidance response, there was a significant difference between groups in the latency to avoid the fish predator (Table 1), where Wild fish (Mdn = 1.5 s, IQR = 2.8) showed an avoidance response significantly faster than Juvenile fish (Mdn = 59.0 s, IQR = 138.0; *p* = 0.011; Figure 5a). Although following a similar trend, there was no significant difference (*p* = 0.08) between groups in time taken to demonstrate an avoidance response to the model avian predator model (Table 1; Figure 5b).

There was a significant difference between groups in the latency to freeze in response to the model avian predator (Table 1; Appendix A), with Soft release fish (Mdn = 2.0 s, IQR = 3.0) freezing significantly more slowly than Hatchery fish (Mdn = 8.0 s, IQR = 8.0; *p* = 0.025). The latency to freeze in response to the fish predator was significant between groups (Table 1), but this was due to the slow response time in Juvenile fish (Appendix A).

### 3.4. Novel Food Test

Of the 118 SPP tested, 29.7% inspected the novel food, with an overall average time of over 23 min to inspection. There was no significant difference between groups in the likelihood of inspecting the food or, for the fish that inspected the food, in the time taken to inspect the novel food (Table 1).

### 3.5. Developmental Context: Juvenile Baseline Data

Given multiple confounding factors (age and testing temperature of 21–22 °C vs. 13–14 °C for adults), comparisons with the Juvenile group are presented for developmental context only. Juvenile fish showed higher activity levels in exploration and habitat choice tests but used ledges significantly less compared to adult fish (Table 1 and Table 2; Figure 3 and Figure 4), likely reflecting combined effects of age and warmer conditions.

## 4. Discussion

Soft-release exposure did not broadly alter emergence, exploration, and novel food investigation, but it offered two clear advantages: (i) a substantial growth/condition advantage (≈14.6 mm and 1.6 g), and (ii) enhanced sheltering behaviour in novel habitats. The differences in size and colouration (Appendix A) indicate that the soft-release site was highly beneficial to the physical development of SPP. The differences between the Soft release and Hatchery and Wild fish might also be due to differences in food availability and diversity as the 100 fish released into our soft-release site was well below the location’s carrying capacity. It is worth noting that Raymond et al. [24] found no difference in SPP body condition between soft-release and wild sites after 3–4 years in the soft-release sites, possibly because the higher number of stocked fish (200) and longer duration had begun to limit body condition. The benefit of a bigger size at wild release may yield better population outcomes due to the reproductive output of larger females [4], and in other species, bigger size at release can improve post-release survival (see, e.g., [28]), which is often linked to bigger fish being less susceptible to predation (e.g., Brown Trout, *Salmo trutta* L. [29]).

In terms of sheltering behaviour, both Soft release and Wild fish used protective ledges significantly more often than Hatchery fish. This pattern suggests that environmental exposure in semi-natural conditions promotes the recognition and use of refuge structures—an ability that aligns with the Southern pygmy perch’s natural preference for dark, structurally complex habitats dominated by aquatic vegetation [30,31]. Such behaviour is ecologically relevant because access to and use of cover can reduce predation risk, lower stress, and improve energy budgets by providing resting sites and reducing the need for constant vigilance. In structurally rich environments, individuals that actively seek and utilise shelter are likely to experience higher survival probabilities, particularly during early post-release stages when predation pressure is typically greatest. Conversely, Hatchery fish, which were reared in clear-water tanks with minimal structural complexity, may have developed a reduced motivation to seek cover because their environment lacked both predators and visual barriers. This familiarity with open, predator-free conditions could explain their lower use of shelter during habitat tests.

We found that, in general, all groups of adult fish showed similar rates of avoidance and freeze responses to fish and model avian predators. However, when we examined the latency to avoid or freeze, we began to see some small differences between groups. First, and perhaps most relevant in our test environment, we found a group effect in latency to avoid the fish (*p* = 0.01) and model bird (*p* = 0.08) predators, with figures suggesting a trend in Wild fish responding faster to the presentation of the predator than other adult fish. In our study, avoidance, which is also called “escape” in the literature, was simply scored as moving away from the predator, but it is important to point out that the precise movements involved (e.g., initial angle of movement, trajectory, speed) are important in the effectiveness of this response [32,33]. The anti-predator behaviour of wild fish may have been shaped through interactions with predators such as redfin perch (*Perca fluviatilis*) in Sheepwash Creek (See Appendix A for fish assemblages). Alternatively, the laboratory testing environment would have been most unfamiliar to the wild fish, and so it may be reasonable that they would show heightened anti-predator behaviour compared to other groups. Indeed, familiarity with the testing conditions has been found to alter anti-predator behaviour in Crimson-spotted rainbowfish (*Melanotaenia duboulayi*) [34]. We suggest that further tests of anti-predator behaviour that examine the behavioural response in more detail, and how it is shaped by predator exposure in more natural conditions, should be undertaken to determine the effect, if any, of the environment on anti-predator behaviour.

Second, and perhaps less relevant as an effective anti-predator response in our laboratory setting, we found that Soft release fish were slower than Hatchery fish to show a freeze response. This finding was unexpected and in opposition to previous findings indicating that fish with experience of wild environments tend to freeze sooner than fish from hatcheries (i.e., in Cod, *Gadus morhua*) [33]. In addition, it is unclear whether showing a freeze response in a test undertaken in clear water with no cover is an effective anti-predator response. In light of these discrepancies and limitations, we suggest that our finding with respect to the freeze response should be interpreted with caution. Nonetheless, the overall similarity in anti-predator behaviour between the Hatchery fish and other adult fish supports the view that anti-predator behaviour in this species has a strong genetic control component [35,36]. Considering this, further genetic and epigenetic investigation, especially in the context of conservation translocations is necessary.

Interestingly, all fish groups had limited approaches and minimal interaction with the novel food. Enriched environments and past exposure to live prey has previously been found to impact willingness to forage novel foods in Atlantic salmon (*Salmo salar*), but only when the novel food was live [37]. It would be useful to undertake further novel food tests using food other than defrosted bloodworm, such as live bait, though the significant difference in the size of the Soft release fish compared to other adult fish indicates that ability to find food in the soft-release site is not stymied by hatchery rearing.

Although our findings suggest that younger fish demonstrated bolder traits than older fish, as shown in giant danio (*Devario aequipinnatus*) [38], it must be remembered that the Juvenile group was tested in warmer water than the other groups, and temperature is known to have a significant impact on fish activity [39]. The water in the home tanks was intentionally kept at a similar temperature to the source water temperature for each fish group to minimise time spent in the hatchery environment. It was judged that this was important to minimise any variation in behaviour resulting from prolonged time in the laboratory environment, but this may have had unintended consequences on the observed differences in activity levels between the Juvenile fish and other groups. Thus, while Juvenile fish tended to display bolder behaviour- moving more in the exploration and habitat choice tests- we cannot rule out the possibility that this was influenced by the higher water temperature of their test environment.

## 5. Conclusions

In this species, the soft-release experience was found to produce advantages in terms of growth, shelter-seeking behaviour, and possibly subtle effects on anti-predator behaviour, though further tests in a more natural environment should be undertaken to confirm ecologically important experience-dependent changes in behaviour. Our findings have important implications for F1 conservation breeding programmes. The minimal behavioural differences between hatchery and soft-release fish suggest that F1 individuals possess sufficient behavioural competence for release without intensive conditioning. Instead, soft-release programmes should prioritise (1) growth and body condition optimisation through access to natural food and reduced competition; (2) opportunities for shelter recognition and possible fine tuning of anti-predator behaviour; and (3) simplified protocols focused on maximising size and survival-relevant traits rather than complex behavioural training. To achieve these changes, we recommend soft-release programmes incorporate submerged vegetation, variable substrates, and live prey to simulate natural foraging and shelter conditions. This approach may offer a cost-effective strategy for enhancing post-release outcomes in F1 fish, especially when wild-type genetics are preserved. However, species-specific validation and field-based survival monitoring remain essential to confirm these laboratory findings.

## Figures and Tables

**Figure 1 animals-15-02754-f001:**
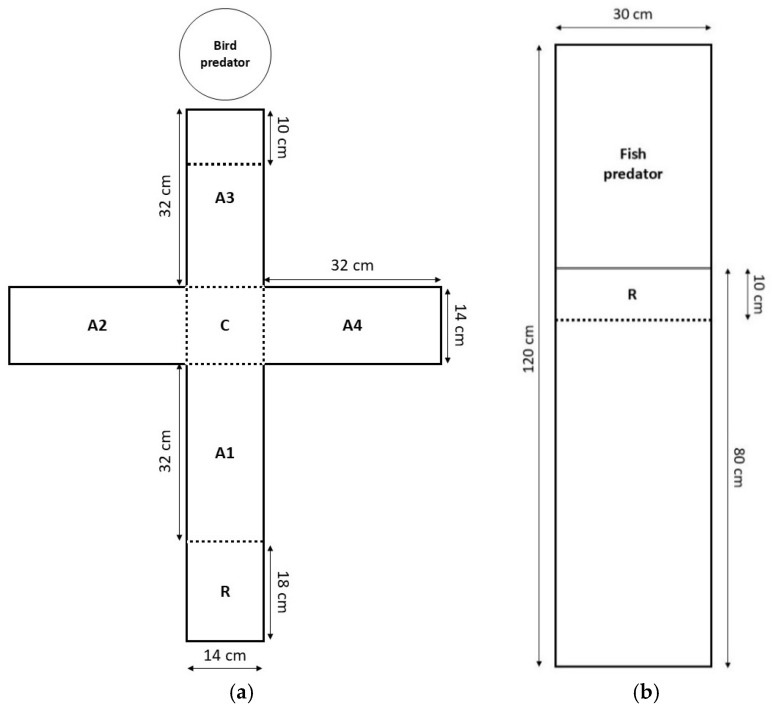
(**a**) Plus maze used for behavioural tests of emergence, exploration, habitat choice, and responses to a model avian predator. Dashed lines indicate placement of removable guillotine doors; refuge for initiating emergence test (R), arms for exploration and habitat placement (A1–A4), and starting location for habitat choice (C). Water depth was at least 30 cm. (**b**) Rectangular tank used for the fish predator test. Dashed lines indicate placement of removable guillotine doors; refuge indicated by “R”, and placement of fish predator behind opaque barrier is also indicated.

**Figure 2 animals-15-02754-f002:**
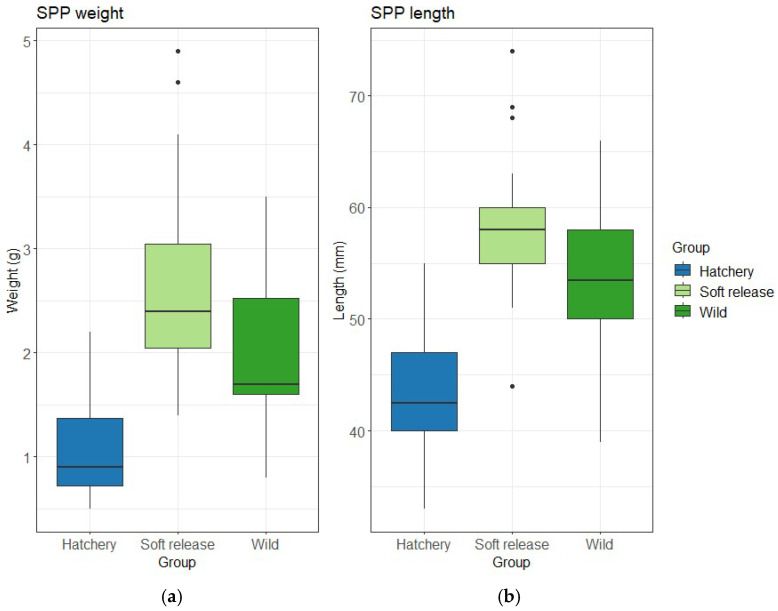
(**a**) Weight and (**b**) length of three groups of adult Southern pygmy perch (SPP). Plots represent mean (line inside boxes) values, 25–75 percent quartiles (boxes), and ranges (“whiskers”). Outliers are represented with a black circle.

**Figure 3 animals-15-02754-f003:**
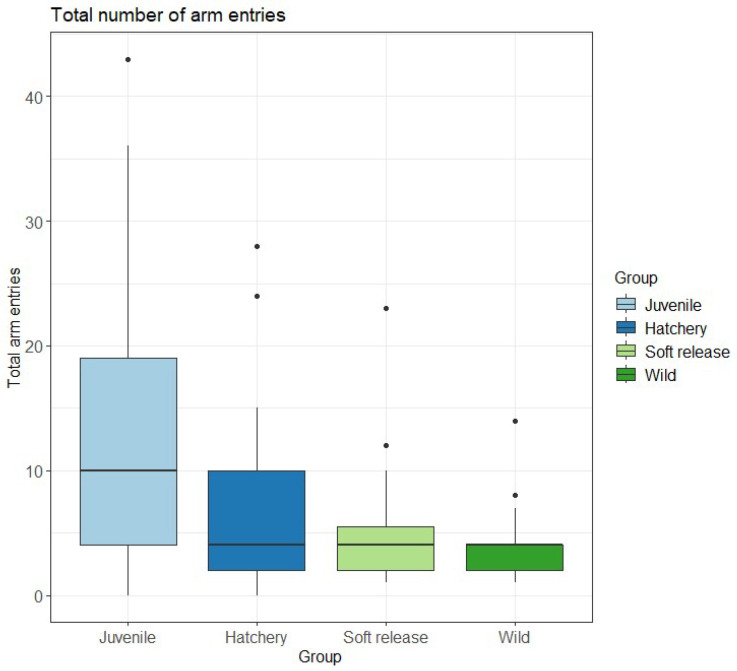
Total number of arms entered in the exploration test by four groups of fish. Plots represent mean (line inside boxes) values, 25–75 percent quartiles (boxes), and ranges (“whiskers”). Outliers are represented with a black circle.

**Figure 4 animals-15-02754-f004:**
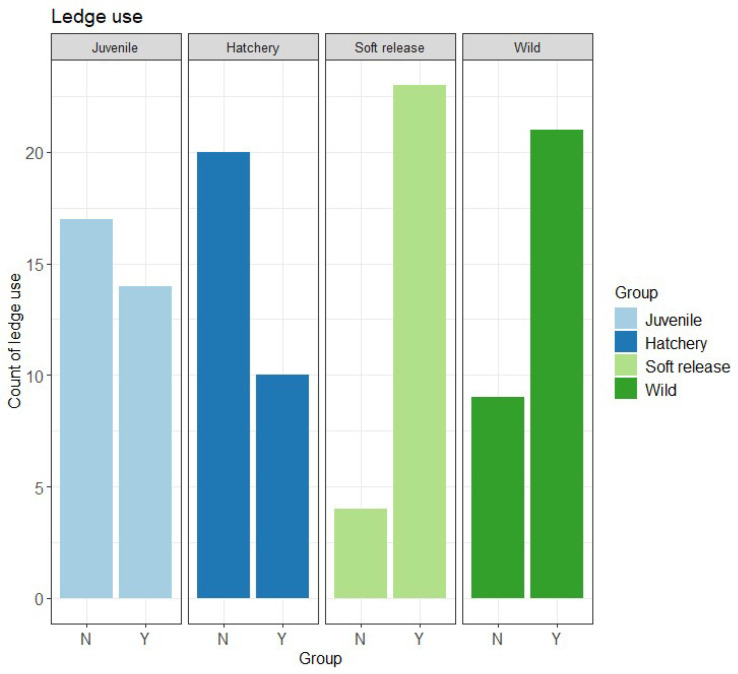
Number of fish using (Y) and not using (N) the ledge in the habitat choice test.

**Figure 5 animals-15-02754-f005:**
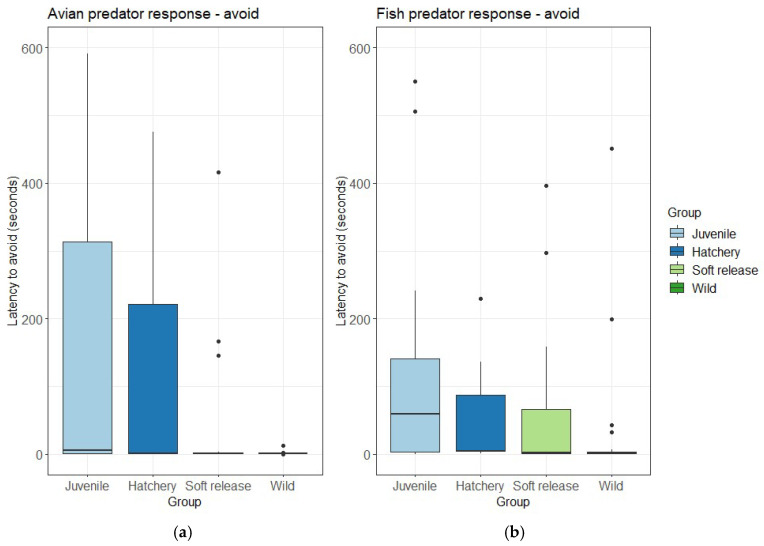
Latency to demonstrate an avoidance response to the (**a**) fish and (**b**) avian predator for fish that demonstrated an avoidance response. Plots represent mean (line inside boxes) values, 25–75 percent quartiles (boxes), and ranges (“whiskers”). Outliers are represented with a black circle.

**Table 1 animals-15-02754-t001:** Summary of statistical tests and pairwise comparisons for behavioural and physical traits.

Behaviour/Test	Global Test (χ^2^/F, df, *p*)	Significant Pairwise Differences (Dunn or GLM)
Size—Weight	KW χ^2^ = 46.465, df = 2, *p* < 0.001	Soft release > Hatchery (*p* < 0.001); Wild > Hatchery (*p* < 0.001); Soft release > Wild (*p* = 0.036)
Size—Length	KW χ^2^ = 42.721, df = 2, *p* < 0.001	Soft release > Hatchery (*p* < 0.001); Wild > Hatchery (*p* < 0.001); Soft release vs. Wild ns (*p* = 0.062)
Emergence latency	KW χ^2^ = 6.132, df = 2, *p* = 0.105	None
Exploration (arms)	KW χ^2^ = 12.281, df = 3, *p* = 0.007	Juvenile > Soft release (*p* = 0.029); Juvenile > Wild (*p* = 0.010)
Ledge use (binary)	GLM F_3,114_ = 20.647, *p* = 0.0001	Wild and Soft release > Hatchery and Juvenile
Avian avoid (likelihood)	GLM F_3,114_ = 3.786, *p* = 0.285	None
Fish avoid (likelihood)	GLM F_3,114_ = 2.101, *p* = 0.551	None
Avian freeze (likelihood)	GLM F_3,114_ = 7.381, *p* = 0.060	None
Fish freeze (likelihood)	GLM F_3,114_ = 1.079, *p* = 0.782	None
Latency—Avian avoid	KW χ^2^ = 6.693, df = 3, *p* = 0.082	None
Latency—Fish avoid	KW χ^2^ = 11.454, df = 3, *p* = 0.010	Wild < Juvenile (*p* = 0.011)
Latency—Avian freeze	KW χ^2^ = 8.992, df = 3, *p* = 0.029	Soft release > Hatchery (*p* = 0.025)
Latency—Fish freeze	KW χ^2^ = 13.832, df = 3, *p* = 0.003	Soft release > Juvenile (*p* = 0.026)
Novel food (inspect)	GLM F_3,114_ = 6.748, *p* = 0.080	None
Novel food (latency)	KW χ^2^ = 1.685, df = 3, *p* = 0.640	None

**Table 2 animals-15-02754-t002:** Percentage of time in each habitat in the habitat choice test. Kruskal–Wallis analysis of habitat choice found no significant difference in habitat use between groups (df = 3).

Group	Empty	Small Rocks	Large Rocks	Plants
Juvenile	6.8%	36.8%	27.9%	28.6%
Hatchery	4.7%	30.7%	22.7%	41.9%
Soft-Release	4.8%	33.3%	28.5%	33.3%
Wild	7.8%	36.6%	21.7%	33.9%
Total	6.1%	34.4%	25.1%	34.4%
KW χ^2^ (df = 3)	1.01	0.38	1.807	1.134
*p*	0.799	0.944	0.613	0.769

## Data Availability

The original data presented in the study are openly available in FigShare at https://doi.org/10.6084/m9.figshare.29985175 (accessed on 16 September 2025).

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
