# Peer review of "Do Hatchery-Reared Southern Pygmy Perch (Nannoperca australis) Develop Effective Survival Behaviour in a Soft-Release Site?"

_animals, 2025, doi:10.3390/ani15182754_

Round 1
Reviewer 1 Report
Comments and Suggestions for Authors
L130-148:Please provide a detailed description of the environmental conditions for the 'Soft release' group of experimental fish during the six-month acclimatization training period. This should include, but not be limited to, water temperature, the biological species present (e.g., cohabiting species, prey, or predators), and the types of shelter or enrichment structures available.
L132:The species name "Nannoperca australis" should be abbreviated to "N. australis" upon its second occurrence in the text, in accordance with standard academic conventions. Please revise this instance and check for similar cases throughout the manuscript to ensure consistency.
L193: "The rationale for setting the acclimation period in the shelter zone to 10 minutes requires clarification. Could you please provide references from previous studies (particularly on similar fish species or experimental setups) that support the adequacy of this specific duration? Alternatively, if this was determined through pre-experimental observations (e.g., confirming that stress behaviors like erratic swimming or freezing plateau within this time), please describe the validation process in the Methods section. A longer acclimation period (e.g., 15-30 minutes) is commonly adopted in behavioral ecology to ensure full habituation to the new environment and stable baseline behavior."
L: 194:Please provide a more detailed description of the experimental procedure. Specifically, clarify the status of the dividing guillotine door in other arms when the dividing guillotine door of Arm A1 is opened. If the protocol involved dynamic control of multiple valves (e.g., sequential or selective opening), please explicitly describe the logic and timing of these operations.
L193-197:It is recommended to conduct a comparative analysis to determine whether there is a statistically significant difference in the subsequent behavioral performance between fish that were "gently ushered out of the refuge area" and those that exited spontaneously. A additional statistical analysis should be performed to evaluate whether this procedural intervention introduced any bias in the behavioral outcomes of the follow-up experiments. If a difference is detected, its potential implications should be discussed. If no difference is found, stating this result would also strengthen the methodological rigor of the study.
L202: Please clarify the specific method used to guide the fish back to the central acclimation zone after the exploration test. For example:“Was the fish gently herded using a non-threatening stimulus (e.g., a slow-moving net or partition)?”. A detailed description of this procedure is essential to ensure the protocol minimizes stress and avoids unintended behavioral biases.
L202: The 5-minute acclimation period in the central zone requires justification. Please provide references supporting this duration for your specific species or similar experimental contexts (e.g., studies using maze assays with comparable fish models). If based on pilot observations, describe the criteria for determining adequacy (e.g., 'stress behaviors like rapid swimming decreased to baseline levels within 5 minutes').
L203: To control for potential spatial bias, please clarify whether the position of the substrate was systematically altered between successive fish trials. If such counterbalancing or randomization was implemented, please specify the method used (e.g., "the substrate was rotated 90°after each trial" or "its location was randomized across predetermined positions"). If no such measures were taken, please acknowledge this as a limitation in the Discussion section and justify why positional effects were deemed negligible for this specific experimental design and species.
L203: Please provide additional details on the particle size range of the stones used in the habitat selection preference experiment, as well as their coverage area and laying thickness within the device. Also, specify the dimensions and quantity of the aquatic plants used.
L204:As mentioned in the text, the basket rim creates a shading effect. Please clarify: if an experimental fish stays near the basket rim in a given substrate, should this be considered valid data in the habitat selection experiment and included in the analysis?
L206: Please provide a clear description of the statistical metric "percentage of residence time in each habitat" used in the habitat selection test. This metric is mentioned in the results section but is not explicitly defined in the methods section.
L201-206:The description of this part of the experimental procedure appears incomplete. The full protocol should involve allowing the experimental fish to acclimate in the central area of the apparatus for 5 minutes, followed by the removal of partitions to open the four maze arms, with behavioral observations then conducted for a period of 10 minutes. It is recommended that the authors revise the Methods section to include these critical temporal and procedural details to ensure the reproducibility of the experiment.
L211:The text "Error! Reference source not found" appears to be an invalid reference marker. Please verify this section and remove the phrase if it does not belong to the main content. Additionally, check the entire document for similar occurrences and correct them accordingly.
Figure1:The manuscript mentions the installation of a dividing guillotine door in the A3 arm zone of the experimental apparatus (e.g., a maze). However, the specific function and rationale for this design choice are not sufficiently detailed. Please clarify the role of this valve.
L217: "The definition of 'freezing behavior' as 'the fish remaining stationary for more than 1 second with no visible movement' requires further support. Is this specific 1-second duration threshold based on established literature? This duration appears to be relatively short for reliably distinguishing freezing from brief pauses in activity. Please provide relevant citations to justify this criterion or consider revising the threshold to align with more conventional standards (e.g., 2 seconds or longer) commonly used in the field."
L217:The term "escape behavior" was used as a behavioral metric in the text, but its operational definition is absent from the Methods section. To ensure clarity and reproducibility of the study, please explicitly describe the specific constellation of behaviors that were defined as "escape behavior" in Southern pygmy perch (Nannoperca australis).
L207-224:The study investigates the responses of experimental fish to different predators. It would be significantly strengthened by providing explicit ecological context regarding the predator-prey relationships between the species used (Nycticorax caledonicus and Maccullochella peelii) and the experimental subject in their natural habitat. Specifically, please clarify whether these two predator species are known to be direct natural predators of the study species.
The rationale for selecting two distinct predator species is not sufficiently justified in the manuscript. Furthermore, it is strongly recommended that the authors perform a comparative analysis of the behavioral responses exhibited by the experimental fish when confronted with these two different predators. A statistical comparison is crucial to determine whether the observed anti-predator behaviors differ significantly based on predator type.
L229:Please specify how the food detection behavior was quantified. For instance, how was it determined whether the experimental fish was within approximately 5 mm of the food?
Please clarify the sample size (n) for each experiment described in this study. Additionally, it is critical to specify whether each individual fish was subjected to all experimental tests sequentially as outlined in the manuscript. If this is the case, it is highly recommended to provide a detailed experimental timeline diagram or flowchart illustrating the sequence and duration of procedures for a single fish.
Please report the exact P-values for all statistical results presented in the manuscript, rather than stating only whether results were significant .
Figure2:Please revise the y-axis title of Figure 2(a) from "Weight (grams)" to "Weight (g)" to ensure consistency with the format used in Figure 2(b).
Figure2:The font size of the figure captions appears insufficient. Please verify whether it complies with the journal's formatting guidelines for publication. If subsequent figures exhibit the same issue, please revise them accordingly.
Table1:The significance indicator "p" should be formatted in italics to conform to standard statistical notation conventions. Please revise this instance and ensure consistent formatting is applied to all occurrences of the significance indicator in the manuscript.
Author Response
L130-148:Please provide a detailed description of the environmental conditions for the 'Soft release' group of experimental fish during the six-month acclimatization training period. This should include, but not be limited to, water temperature, the biological species present (e.g., cohabiting species, prey, or predators), and the types of shelter or enrichment structures available.
Thank you and we have provided a more detailed description of the pond and vegetation, as these were the only additional details we gathered.
L132:The species name "Nannoperca australis" should be abbreviated to "N. australis" upon its second occurrence in the text, in accordance with standard academic conventions. Please revise this instance and check for similar cases throughout the manuscript to ensure consistency.
Thank you. We have changed as suggested, with the exception that we left the complete name in the simple summary and abstract, since these parts may be read separately.
L193: "The rationale for setting the acclimation period in the shelter zone to 10 minutes requires clarification. Could you please provide references from previous studies (particularly on similar fish species or experimental setups) that support the adequacy of this specific duration? Alternatively, if this was determined through pre-experimental observations (e.g., confirming that stress behaviors like erratic swimming or freezing plateau within this time), please describe the validation process in the Methods section. A longer acclimation period (e.g., 15-30 minutes) is commonly adopted in behavioral ecology to ensure full habituation to the new environment and stable baseline behavior."
Thank you for these comments. We have added a couple of sentences to explain that the acclimating periods were sufficient for behaviour to return to normal. There are several reasons for this. First, the water in the testing tanks was connected to the home tanks, so there was no effect of “shock” of moving to different water that other researchers may have to deal with. Second, the tanks were in the room next to the holding tanks, so the netting period was extremely short (less than 3 seconds usually). Lastly, we have used these methods with several fish species, and their behaviour also returns to normal with these brief acclimating periods (e.g. Freire, R., Michie, M., Rogers, L. and Shamsi, S. (2024). Age-related changes in survival behaviour in parasite-free hatchery-reared Rainbow trout (Oncorhynchus mykiss). Animals, 14, 1315.; Freire, R., Rogers, L., Creece, D. and Shamsi, S. (2022). Neophobic behavioural responses of parasitised fish to a potential predator and baited hook. Applied Animal Behaviour Science, 254: 105722; Shamsi, S., Rogers, L., Sales, E., Kopf, R.K. and Freire, R. (2021). Do parasites influence behavioural traits of wild and hatchery-reared Murray cod, Maccullochella peeli? Parasitology Research, 1-9.)
L: 194:Please provide a more detailed description of the experimental procedure. Specifically, clarify the status of the dividing guillotine door in other arms when the dividing guillotine door of Arm A1 is opened. If the protocol involved dynamic control of multiple valves (e.g., sequential or selective opening), please explicitly describe the logic and timing of these operations.
Thank you. We have added to the text to describe which arms were blocked with guillotine doors for each test.
L193-197:It is recommended to conduct a comparative analysis to determine whether there is a statistically significant difference in the subsequent behavioral performance between fish that were "gently ushered out of the refuge area" and those that exited spontaneously. A additional statistical analysis should be performed to evaluate whether this procedural intervention introduced any bias in the behavioral outcomes of the follow-up experiments. If a difference is detected, its potential implications should be discussed. If no difference is found, stating this result would also strengthen the methodological rigor of the study.
Thank you and yes, we too considered this. However, only 9 out of 118 fish failed to exit the refuge within the time limit (2 Juvenile, 2 Hatchery, 1 Soft release and 4 Wild fish). The small number of fish that failed to exit from each treatment was too small for meaningful statistical analysis.
L202: Please clarify the specific method used to guide the fish back to the central acclimation zone after the exploration test. For example:“Was the fish gently herded using a non-threatening stimulus (e.g., a slow-moving net or partition)?”. A detailed description of this procedure is essential to ensure the protocol minimizes stress and avoids unintended behavioral biases.
Thank you, we have added that we used the handle of the net to usher these fish.
L202: The 5-minute acclimation period in the central zone requires justification. Please provide references supporting this duration for your specific species or similar experimental contexts (e.g., studies using maze assays with comparable fish models). If based on pilot observations, describe the criteria for determining adequacy (e.g., 'stress behaviors like rapid swimming decreased to baseline levels within 5 minutes').
Thank you and as mentioned above in response to the acclimation period after moving, we based this on behaviour returning to normal and provided some references.
L203: To control for potential spatial bias, please clarify whether the position of the substrate was systematically altered between successive fish trials. If such counterbalancing or randomization was implemented, please specify the method used (e.g., "the substrate was rotated 90°after each trial" or "its location was randomized across predetermined positions"). If no such measures were taken, please acknowledge this as a limitation in the Discussion section and justify why positional effects were deemed negligible for this specific experimental design and species.
Thank you and we agree completely that spatial cues can be extremely important. For this reason the placement of each substrate was randomised for each fish. This is mentioned in section 2.3, and we have added the “predetermine positions” as suggested above to further clarify this point.
L203: Please provide additional details on the particle size range of the stones used in the habitat selection preference experiment, as well as their coverage area and laying thickness within the device. Also, specify the dimensions and quantity of the aquatic plants used.
Thank you. We have added stone size and that the plants were artificial. A photo of these substrates is provided in the Supplementary material, Figure S2b.
L204:As mentioned in the text, the basket rim creates a shading effect. Please clarify: if an experimental fish stays near the basket rim in a given substrate, should this be considered valid data in the habitat selection experiment and included in the analysis?
Thank you and we think such analysis would not reveal much. When a fish was “under a ledge” it was just beside the basket with substrate, so was neither above nor could see the substrate since the side of the ledge was solid. “Under the ledge” was therefore a very similar environment irrespective of which substrate was in the basket. For this reason we considered that a binomial analysis (i.e. used ledge/did not use ledge) was a sensible way to analyse this unexpected behavioural response.
L206: Please provide a clear description of the statistical metric "percentage of residence time in each habitat" used in the habitat selection test. This metric is mentioned in the results section but is not explicitly defined in the methods section.
Thank you, we have added a sentence to section 2.6 to explain that counts of the number of times fish were recorded in each habitat were converted to percentages.
L201-206:The description of this part of the experimental procedure appears incomplete. The full protocol should involve allowing the experimental fish to acclimate in the central area of the apparatus for 5 minutes, followed by the removal of partitions to open the four maze arms, with behavioral observations then conducted for a period of 10 minutes. It is recommended that the authors revise the Methods section to include these critical temporal and procedural details to ensure the reproducibility of the experiment.
Thank you, we have added further details to this paragraph to more fully describe our method.
L211:The text "Error! Reference source not found" appears to be an invalid reference marker. Please verify this section and remove the phrase if it does not belong to the main content. Additionally, check the entire document for similar occurrences and correct them accordingly.
Thank you, there appears to have been a glitch in reproduction, which we have now corrected and checked and corrected other glitches.
Figure1:The manuscript mentions the installation of a dividing guillotine door in the A3 arm zone of the experimental apparatus (e.g., a maze). However, the specific function and rationale for this design choice are not sufficiently detailed. Please clarify the role of this valve.
Thank you. We have now added to section 2.4 that this area was where the fish was placed to acclimate prior to the model bird predator test.
L217: "The definition of 'freezing behavior' as 'the fish remaining stationary for more than 1 second with no visible movement' requires further support. Is this specific 1-second duration threshold based on established literature? This duration appears to be relatively short for reliably distinguishing freezing from brief pauses in activity. Please provide relevant citations to justify this criterion or consider revising the threshold to align with more conventional standards (e.g., 2 seconds or longer) commonly used in the field."
Thank you for this suggestion. To our knowledge no one has undertaken these types of behavioural test with this species, so we based our method on our experience of this test with other species. In this particular species, fin movement was very noticeable as was a freezing of fin movement following presentation of the predator. Fish typically resumed movement or moved away after some time, so a longer threshold would not have yielded a different outcome. We therefore chose a threshold of 1 sec which we also consider more ecologically relevant than anything longer (ie. A freeze response is not very effective if the predator sees prior movement).
L217:The term "escape behavior" was used as a behavioral metric in the text, but its operational definition is absent from the Methods section. To ensure clarity and reproducibility of the study, please explicitly describe the specific constellation of behaviors that were defined as "escape behavior" in Southern pygmy perch (Nannoperca australis).
Thank you for this comment. We have done a search of our document, and found that we only used “escape behaviour” once. This was in the discussion where we discuss the similarities and differences of escape behaviour with our metric of avoidance. We noticed though that in the method we had used “moved away” rather than “avoidance” so have now rectified this to use the same term throughout.
L207-224:The study investigates the responses of experimental fish to different predators. It would be significantly strengthened by providing explicit ecological context regarding the predator-prey relationships between the species used (Nycticorax caledonicus and Maccullochella peelii) and the experimental subject in their natural habitat. Specifically, please clarify whether these two predator species are known to be direct natural predators of the study species.
Thank you and unfortunately little is known about predators of this species. However, Murray cod is a common apex predator in these systems, and take all small fish. Also, it has been reported that birds threaten SPP (Pearce, L. (2014). Conservation management of southern pygmy perch (Nannoperca australis) in NSW, in the context of climactic extremes and alien species.) In the absence of clear evidence of predation of SPP by these predators, merely an assumption that they do, we think it is perhaps sufficient to focus on the behavioural responses which clearly showed a response to these potential predators.
The rationale for selecting two distinct predator species is not sufficiently justified in the manuscript. Furthermore, it is strongly recommended that the authors perform a comparative analysis of the behavioral responses exhibited by the experimental fish when confronted with these two different predators. A statistical comparison is crucial to determine whether the observed anti-predator behaviors differ significantly based on predator type.
Thank you and yes, we did explore this possibility. However, due to the size of the predator fish the fish predator test was undertaken in a larger tank than the model bird predator test. Therefore, predator type would have been confounded by tank size reducing the value of any direct statistical comparison. We agree that readers would be interested in the comparison and for that reason we have presented responses to both predators on the same figure (with figure a and b) to allow readers to at least visually compare the responses.
L229:Please specify how the food detection behavior was quantified. For instance, how was it determined whether the experimental fish was within approximately 5 mm of the food?
Thank you. This was based on an estimate by the experimenter watching the fish. We have now included this in section 2.5.
Please clarify the sample size (n) for each experiment described in this study. Additionally, it is critical to specify whether each individual fish was subjected to all experimental tests sequentially as outlined in the manuscript. If this is the case, it is highly recommended to provide a detailed experimental timeline diagram or flowchart illustrating the sequence and duration of procedures for a single fish.
We agree that timelines can be useful with complex experimental sequences. However, our experiment was quite straightforward; all 118 received all tests in the order described. We have added “on all fish in this order” to a sentence in section 2.3 to make this clear. We think that in the interest of brevity the explanation provide in section 2.3 is sufficient.
Please report the exact P-values for all statistical results presented in the manuscript, rather than stating only whether results were significant .
We have provided exact p values for all probabilities above 0.001. When p<0.001, we have written that as P<0.001. This is to avoid providing figures that have a large number of zeros, and in our opinion don’t’ tell the reader much more than P<0.001.
Figure2:Please revise the y-axis title of Figure 2(a) from "Weight (grams)" to "Weight (g)" to ensure consistency with the format used in Figure 2(b).
Thank you, changed as suggested
Figure2:The font size of the figure captions appears insufficient. Please verify whether it complies with the journal's formatting guidelines for publication. If subsequent figures exhibit the same issue, please revise them accordingly.
Thank you, we have increased the font size on all figures.
Table1:The significance indicator "p" should be formatted in italics to conform to standard statistical notation conventions. Please revise this instance and ensure consistent formatting is applied to all occurrences of the significance indicator in the manuscript.
Thank you, p now in italics throughout
Reviewer 2 Report
Comments and Suggestions for Authors
Thanks for providing me the opportunity to review the paper titled “Do hatchery-reared Southern pygmy perch (Nannoperca australis) develop effective survival behaviour in a soft release site?”. This study provide a valuable information to conservation biology by empirically testing the behavioural plasticity of captive-bred Southern pygmy perch under soft-release conditions a topic often underrepresented in freshwater fish recovery programs. The use of multiple behavioural assays (emergence, exploration, habitat choice, predator response, and novel food tests) provides a robust framework for assessing phenotype development and ecological readiness.
I have few suggestions if considered may improve the manuscript
- While the laboratory-based behavioural tests are informative, they may not fully capture the complexity of survival dynamics in natural ecosystems. Field-based validation is essential to confirm the ecological relevance of these traits. The suggestion that survival behaviour is largely inherited rather than learned warrants further genetic and epigenetic investigation, especially in the context of conservation translocations.
- Rewrite. Line number 12-13 “Captive breeding and release of threatened small-bodied freshwater fish is a common conservation method…”My suggestion is consider specifying the geographic or ecological context (e.g., Australian freshwater systems) to anchor the relevance. You can write alternative: “Captive breeding and release programs are widely used to conserve threatened small-bodied freshwater fish, particularly in temperate Australian ecosystems…”
- Clarify “wild phenotype”. The term “wild phenotype” is central but vague. Consider briefly defining what behavioral traits constitute this phenotype in the context of the study.
- The mention of size differences (14.6 mm, 1.6 g) is useful, but it would be stronger if framed with statistical significance (e.g., “significantly larger, p < 0.05”).
- “Possibly showed altered anti-predator responses” is too tentative. If the data were inconclusive, consider stating the limitations more directly. My suggestion is that “While trends suggested altered anti-predator responses, statistical support was limited, warranting further investigation.”
- The final recommendation is valuable but could be more actionable. For instance, what specific structural elements or foraging setups are most beneficial? “We recommend soft-release programs incorporate submerged vegetation, variable substrate, and live prey to simulate natural foraging and shelter conditions…”
- Some sentences are long and packed with multiple ideas. Breaking them up would improve readability. Example: “We conclude that in this species the soft-release experience has advantages on growth, shelter-seeking and possibly subtle effects on predator avoidance…” suggested “We conclude that soft-release experience enhances growth and shelter-seeking behavior in this species. Subtle effects on predator avoidance were also observed, though further testing is needed…”
- Consider emphasizing the need for field-based validation to bridge lab findings with real world survival outcomes. Behavioral plasticity vs. inheritance: The conclusion that survival behavior is “mostly inherited” could benefit from a brief mention of how this was inferred e.g., lack of change despite environmental exposure.
Author Response
Thanks for providing me the opportunity to review the paper titled “Do hatchery-reared Southern pygmy perch (Nannoperca australis) develop effective survival behaviour in a soft release site?”. This study provide a valuable information to conservation biology by empirically testing the behavioural plasticity of captive-bred Southern pygmy perch under soft-release conditions a topic often underrepresented in freshwater fish recovery programs. The use of multiple behavioural assays (emergence, exploration, habitat choice, predator response, and novel food tests) provides a robust framework for assessing phenotype development and ecological readiness.
I have few suggestions if considered may improve the manuscript
1. While the laboratory-based behavioural tests are informative, they may not fully capture the complexity of survival dynamics in natural ecosystems. Field-based validation is essential to confirm the ecological relevance of these traits. The suggestion that survival behaviour is largely inherited rather than learned warrants further genetic and epigenetic investigation, especially in the context of conservation translocations.
Thank you for this comment and we agree and mention the ecological relevance in the last line of the conclusion and second to last sentence of the abstract. We agree though that discussion of the need for more genetic analysis could be more explicit, and have added the following sentence to the end of the paragraph in the discussion on this topic : “Considering this, further genetic and epigenetic investigation, especially in the context of conservation translocations is necessary.”
2. Rewrite. Line number 12-13 “Captive breeding and release of threatened small-bodied freshwater fish is a common conservation method…”My suggestion is consider specifying the geographic or ecological context (e.g., Australian freshwater systems) to anchor the relevance. You can write alternative: “Captive breeding and release programs are widely used to conserve threatened small-bodied freshwater fish, particularly in temperate Australian ecosystems…”
Thank you: we had changed to include the second suggested text.
3. Clarify “wild phenotype”. The term “wild phenotype” is central but vague. Consider briefly defining what behavioral traits constitute this phenotype in the context of the study.
Thank you and we agree this term is vague, and doesn’t fit well with the work on behaviour. We have therefore used the more precise “natural behaviour” term, and added a sentence to define what we mean by this term to the introduction: “Here, we consider natural behaviour as innate and learned actions and responses that an animal would typically exhibit in its native, wild environment”.
4. The mention of size differences (14.6 mm, 1.6 g) is useful, but it would be stronger if framed with statistical significance (e.g., “significantly larger, p < 0.05”).
“both p<0.001” added and full statistical output provide din the table.
5. “Possibly showed altered anti-predator responses” is too tentative. If the data were inconclusive, consider stating the limitations more directly. My suggestion is that “While trends suggested altered anti-predator responses, statistical support was limited, warranting further investigation.”
Thank you, we have changed as suggested.
6. The final recommendation is valuable but could be more actionable. For instance, what specific structural elements or foraging setups are most beneficial? “We recommend soft-release programs incorporate submerged vegetation, variable substrate, and live prey to simulate natural foraging and shelter conditions…”
Thank you and we agree a statement such as this would be helpful. However, rather than adding it to the abstract, we have added the following to the conclusion “To achieve these changes, we recommend soft release programs incorporate submerged vegetation, variable substrate and live prey to simulate natural foraging and shelter conditions.” We think this is a better place for this suggestion as we would like to see these actions being field tested prior to implementation.
7. Some sentences are long and packed with multiple ideas. Breaking them up would improve readability. Example: “We conclude that in this species the soft-release experience has advantages on growth, shelter-seeking and possibly subtle effects on predator avoidance…” suggested “We conclude that soft-release experience enhances growth and shelter-seeking behavior in this species. Subtle effects on predator avoidance were also observed, though further testing is needed…”
Sentence changed as suggested as well as a similar complex sentence in the simple summary.
8. Consider emphasizing the need for field-based validation to bridge lab findings with real world survival outcomes. Behavioral plasticity vs. inheritance: The conclusion that survival behavior is “mostly inherited” could benefit from a brief mention of how this was inferred e.g., lack of change despite environmental exposure.
We have split the complex sentence at the end of the abstract in two to highlight this point, and replace with a simpler sentence as suggested above.
Reviewer 3 Report
Comments and Suggestions for Authors
Dear authors, please see the attached file.

Author Response
General comments
This manuscript is interesting and explores the effects of six months of soft-release exposure on captive-bred, first-generation Southern pygmy perch Nannoperca australis Günther, 1861. The study combines behavioural (emergence, exploration, habitat choice), predator, and novel food testing to investigate whether F1 Southern pygmy perch developed a wild phenotype.
Specific comments
Introduction section:
I always recommend that authors credit taxonomists properly when mentioning a species for the first time in a manuscript. Please include the taxonomist's name (Günther, 1861), for example, as in the case of the Southern pygmy perch, Nannoperca australis Günther, 1861. It would be appreciated if you could review the manuscript and apply this to other species as well. Thank you.
Thank you for this comment and we agree that for articles focussing on taxonomy, this information would indeed be important. Since this paper focusses on behaviour, we suggest that for the sake of simplifying the paper and maintaining flow we don’t provide this information in this paper.
As per Animals’ instructions for authors, references must be numbered in order of appearance in the text (including table captions and figure legends) and listed individually at the end of the manuscript. Please replace the references found in the main text with numbers (e.g., Lintermans et al., 2020 should be replaced with [1], and so on).
Yes thank you. The original paper followed Animals’ “free-format” guidelines, but we agree that since this paper is now further along in production we should format according to the Journal’s guidelines. We have now changed to the correct referencing style, as well as adhering to all other formatting requirements.
Lines 110-115: I suggest that the authors include, in addition to the references where Southern pygmy perch is listed as endangered or vulnerable, the IUCN Red List of Threatened Species reference https://www.iucnredlist.org/species/123358579/123382811, where it is listed as near threatened.
Thank you. We have added mention of the IUCN red list. (We had already cited the page so simply altered the sentence).
Materials and Methods section:
In section 2.1. Subjects and source of fish, I suggest that the authors include the mean wet body weight and total length of the fish when describing the analyzed groups, for a better understanding of the morphological traits of the specimens used, either in the text or as a table/figure (e.g., “Hatchery” fish (N = 30; 17 females with a wet body weight of X ± sx and total length X ± sx, and 13 males with a wet body weight of X ± sx and total length X ± sx)).
Thank you for this suggestion and we agree that providing some information on fish size at this point will help the reader understand the suitability of the test tanks. In the interest of brevity, we have provided the length of fish from the three treatments in the brackets as suggested. In this species there is little size variation between the sexes (the largest mean difference was 3 mm) so we have not provided measures for each sex. Lastly, figures and statistical results of length and size differences are provided in the results section. I hope our approach is a good compromise between providing this information as requested and maintaining brevity and readability.
Also, since Wild and Soft-release fish sampling was performed via backpack electrofishing, I recommend that the authors describe the setup and apparatus used.
Thank you for pointing out this omission. We have now included details of the backpacking equipment and method, and also describe the bait traps we used.
Lines 210-211: Please verify the S2cError! Reference source not found, and if necessary, remove it.
Thank you. The text has been removed and the correct figure now cited.
Lines 231-234: Why were the adult fish euthanised?
Thanks. We euthanised the fish to examine them for parasites, which is being reported in another paper.
Section 2.6. Statistical analysis should be reorganized to make the statistical approach easier to follow. For example, the test used for the statistical procedure to determine if the dataset comes from a normal distribution or not should be mentioned (e.g., Shapiro-Wilk or Kolmogorov-Smirnov (K-S) tests).
Thank you. Details of normality tests added and complex sentence split into two.
Results section:
Lines 252-253: “Scheme 1. 6 g; Table 1; Figure 2a) and length (mean difference = 14.59 mm; Table 1; 252 Figure 2b), representing approximately 20–30% size advantages”. This paragraph should be verified and corrected.
Thank you. There appeared to be a few of these glitches arising from reproduction. We have corrected this one and checked the manuscript for others.
As mentioned earlier, the mean wet body weight and total length of the fish should be introduced in the Materials and Methods section. Therefore, I have a question regarding Figure 2, where the wet body weight and length of three groups of adult SPP are presented. At what stage of the experiment was it determined?
Thank you, we have now added that euthanasia occurred immediately after the last behavioural test.
Also, since the authors mention the differences in size as an important advantage for soft release, based on these measurements, the Length-Weight Relationship (LWR), Relative Condition Factor (Kn), and Fulton Condition Factor (K) could easily have been calculated to better analyze fish growth, condition, and population dynamics.
Thank you for this suggestion. We have now calculated and provide Fulton condition scores for these three groups, and provide a reference to this metric in the methods.
In addition, I suggest that for each Figure, a legend should be inserted to better describe its content. I am not sure if this applies to authors' figures; please verify the following example: Plots represent median (line inside boxes) values, 25–75 percent quartiles (boxes), and minimal and maximal values shown with vertical lines (“whiskers”). Outliers are represented with a black circle.”
Thank you. We have added an explanation to each figure to make this clear.
Discussion section:
Lines 313-316: The authors mention the differences in size as an important advantage for soft release, but in the manuscript, it’s not clear to the reader, either in the Materials and Methods or Results sections, what the recorded values were. Please clarify this further. Thank you.
Mean lengths, weights and Fulton K scores are now provide in the results.
Line 357: The most commonly used common name for Melanotaenia duboulayi is Crimson-spotted rainbowfish. Please verify and replace if appropriate.
Name verified and corrected.
Lines 422-436: Please insert the Supplementary files link as per Animals Journal Instructions for Authors: “The following are available online at …., Table S1: Latency to emerge from the refuge (mean, median, and IQR); Table S2: Percentage of fish demonstrating particular responses in the predator tests. Note that a fish may have demonstrated both an avoid and freeze response; Table S3: Latency to food inspection (mean, median, and IQR) in the novel food test; Table S4: Summary of fish assemblages from Sheepwash Creek (site of Wild fish) from surveys using fyke nets and bait traps deployed in 2019, 2022, 2023 and 2024; Figure S1: Collection points showing Soft Release fish collection (SR) from a man-made pond, and Wild fish collection (W) from a creek system; etc.”
Thank you, we have now inserted this text as per the author guidelines.
References section:
This section should be formatted as per the Animals Journal Instructions for Authors. Thank you
Thank you, we have now formatted the references according to the guidelines.
Round 2
Reviewer 2 Report
Comments and Suggestions for Authors
The authors addressed my comments, hence recommend to publish
Reviewer 3 Report
Comments and Suggestions for Authors
The work has been significantly revised and is now clearer and more logically organized.